# Collagen-Based Biomaterials in Periodontal Regeneration: Current Applications and Future Perspectives of Plant-Based Collagen

**DOI:** 10.3390/biomimetics7020034

**Published:** 2022-03-24

**Authors:** Thunwa Binlateh, Peungchaleoy Thammanichanon, Pawornwan Rittipakorn, Natthapol Thinsathid, Paiboon Jitprasertwong

**Affiliations:** 1Institute of Research and Development, Suranaree University of Technology, Nakhon Ratchasima 30000, Thailand; thunwa.bin@g.sut.ac.th; 2Institute of Dentistry, Suranaree University of Technology, Nakhon Ratchasima 30000, Thailand; chaleoy@sut.ac.th (P.T.); r.paw@sut.ac.th (P.R.); natthapol@sut.ac.th (N.T.)

**Keywords:** periodontal regeneration, collagen, hemostasis, wound healing, guided bone regeneration, gene regenerative therapy

## Abstract

Collagen is the most widely distributed protein in human body. Within the field of tissue engineering and regenerative medical applications, collagen-based biomaterials have been extensively growing over the past decades. The focus of this review is mainly on periodontal regeneration. Currently, multiple innovations of collagen-based biomaterials have evolved, from hemostatic collagen sponges to bone/tissue regenerative scaffolds and injectable collagen matrices for gene or cell regenerative therapy. Collagen sources also differ from animal to marine and plant-extracted recombinant human type I collagen (rhCOL1). Animal-derived collagen has a number of substantiated concerns such as pathogenic contamination and transmission and immunogenicity, and rhCOL1 is a potential solution to those aforementioned issues. This review presents a brief overview of periodontal regeneration. Also, current applications of collagen-based biomaterials and their mechanisms for periodontal regeneration are provided. Finally, special attention is paid to mechanical, chemical, and biological properties of rhCOL1 in pre-clinical and clinical studies, and its future perspectives in periodontal regeneration are discussed.

## 1. Introduction

Periodontal regeneration involves multi-disciplinary approaches towards re-establishment of periodontal ligament, cementum, and alveolar bone surrounding teeth. Success in periodontal regeneration requires the suitable niches for neovascularization, adequate signal molecules, and proliferation and differentiation of the cells involving regeneration [1,2]. Several biomaterials, technologies, and approaches have been identified to produce the favorable environments for regeneration. During the past decade, numerous innovations have occurred in the field of collagen-based biomaterials. Hemostatic collagen sponges, bone/tissue regenerative scaffolds, and injectable collagen matrices for gene or cell regenerative therapy have been developed and improved for regenerative medical applications, including periodontal regeneration. Not only are these innovative collagen-based biomaterials, but the sources of collagen have also differed from animal to marine and plant-derived recombinant human type I collagen (rhCOL1) sources. Therefore, the aim of this review was to present a brief introduction of periodontal regeneration, followed by current applications of collagen-based biomaterials and their underlying mechanism of action. Ultimately, the recent advances of recombinant expression system in plants that have been evolved for a viable source of collagen and its future perspectives in periodontal regeneration will be provide and discussed.

## 2. Concept of Regeneration in Periodontics

The term regeneration is described as a therapeutic approach involving the replacement or regeneration of damaged human cells and tissues, including organs, to re-establish and restore their normal physiological functions [3]. Regenerative medicine is a translational research integrating a variety of disciplines, such as cell biology, polymer sciences, material sciences, chemistry, medicine, nanotechnology, and manufacturing [4]. In dental practices, this treatment strategy has a long history of use, particularly in periodontal diseases. Tissue regeneration generally requires stemness properties of the tissue. Periodontal tissue has been demonstrated to present stem cells in tissue surrounding teeth, called periodontal ligament (PDL) cells, that possibly involve regeneration of periodontal tissue [5]. PDL is composed of multiple components, including type I collagen fiber bundles, fibroblasts, a group of bone cells found on the surface of alveolar bone, cementoblasts on cementum surface, and mesenchymal stem cells (MSC) [6].

Regeneration requires at least three essential processes: new blood supply or angiogenesis, adequate molecular signaling pathways, and proliferation and differentiation towards the cells that are able to regenerating the damaged tissues [1,2]. During regeneration of periodontal tissue, having sufficient blood supply is extensively needed, since signal molecules that are involved in regeneration will arrive via blood vessels. MSC initially exerts angiogenic effect, which stimulates early formation of granulation tissue and is followed by new vascular network formation [1]. Several angiogenic factors have been revealed in MSC, including IL-8, insulin-like growth factor 1 (IGF-1), monocyte chemoattractant protein 1 (MCP-1), platelet-derived growth factor AA (PDGF-AA), and vascular endothelial growth factor (VEGF) [7]. These factors induce migration of endothelial lineage for neovascularization [8]. In addition, some studies have revealed trans-differentiation of MSC into distinct phenotypes, including endothelial lineage [9].

After neovascular formation, a variety of necessary signals, such as IL-6, IGF-1, TNF-α, and transforming growth factor β1 (TGF-1β), and cells, including monocytes and lymphocytes can be identified. It is known that TNF-α involves the differentiation of osteoblasts by increased mineralization-related genes (bone morphogenic protein 2; BMP2, runt-related transcription factor 2; RUNX2, alkaline phosphatase; ALP) and mineralization of ECM through NF-κB activation [10]. Meanwhile, IL-6 promotes osteocalcin production in osteoblasts, causing amplification of BMP2 and RUNX2 ligands for synthesizing and generating ECM via JAK/STAT3 signaling pathway. IL-6 also augments osteocalcin production and ALP activity though mTOR pathway [1]. In addition to osteogenic differentiation, MSC is implicated in the remodeling of ECM by inhibited metalloproteinase (MMP)-1 and 8 and increased tissue inhibitors of metalloproteinases (TIMPs) using TGF-1β signal [1,11].

On the other hand, secreted IGF-1 contributes to differentiation of fibroblasts and cementoblasts, participating in the secretion of collagen 1 (COL-1) and cementum protein 1 (CEMP-1), respectively, thereby forming collagen fiber and synthesizing radicular cementum at the lesion area of periodontal tissue [1].

Furthermore, the traveling of monocytes and lymphocytes into the area of lesion via new blood vessels is implicated in negative regulation of osteoclast activity in balancing homeostasis of bone remodeling during bone formation. Both cells produce IL-10 that amplifies the levels of osteoprotegerin (OPG), but suppresses expression levels of receptor activator of nuclear factor kappa beta (RANKL), nuclear factor of activated T-cell cytoplasmic 1 (NFATc1), and macrophage colony-stimulating factor (M-CSF) [12]. As a result, the differentiation and activation of osteoclast are inhibited, thus preventing bone resorption (Figure 1).

In spite of the fact that mesenchymal stem cells in PDL are able to differentiate into fibroblasts, osteoblasts, and cementoblasts [1,10,11], the damaged periodontal tissue does not repair successfully through endogenous regenerative processes after conventional treatment being performed. An endogenous regeneration often results from healing by long junctional epithelial formation [13]. Therefore, it is necessary to explore and develop methods that can induce proper proliferation and differentiation of PDL stem cells with additional capability in protecting soft tissue ingrowth.

## 3. Current Applications, Mechanism of Action, and Limitations of Collagen-Based Biomaterials in Periodontal Regeneration

The main applications of collagen are as follows (Table 1): collagen plugs/sponges used for being hemostatic materials in the control of bleeding; resorbable collagens applied in oral wound dressing, acceleration of wound healing, and closure for graft and extraction site; and collagen membranes used as barriers to protect epithelial migration/ingrowth for allowing regenerative ability of pluripotent cells at area of lesion to repopulate in periodontal and implant treatments (Figure 2).

### 3.1. Hemostasis and Wound Healing

Collagen, particularly in type I collagen, is used for oral wound protection with prevented bleeding from the wound, and accelerated endogenous wound healing process [30,31]. Several types and forms of collagen-based products are available, including sponge-like structure, plug-like structure, microfibrillar forms, matrices, scaffold, and drug delivery carrier [30,32,33]. Clinical-based evidence suggests a positive outcome of collagen products. Application of collagen plugs in the third molar extraction resulted in relatively reduced complication rates in comparison to general conditions of previous studies [26,28]. Collagen plugs (CollaPlug^®^) are used to fill extraction wounds, enhance clot formation, accelerate formation of granulation tissue, and reduce postoperative swelling and pain [26]. Ateloplug, a 100% absorbable collagen sponge, appeared to provide hemostasis when applied to the extraction socket. In addition, collagen-containing dressing showed faster wound healing by deposition of fiber in granulation tissue, accelerated re-epithelization, and promoted remodeling of ECM [27,28].

After being placed, collagen rapidly absorbs blood by generating artificial clot-like structure to stop bleeding, thereby reducing swelling and pain postoperatively. Collagen affects coagulation processes. Once it contacts the blood, collagen provides a three-dimensional matrix via binding with a large number of platelets, causing platelet aggregation. Aggregated platelet is granulated to secrete thromboxane A2 that helps and strengthens the formation of blood clot [34,35]. Later, the collagen degrades and collagen fragments are released to act as mediators of inflammation by recruiting neutrophils and macrophage, and enhancing immune responses [36,37]. One study indicated that the inflammatory process induced by collagen is a robust and sharp response, which is transient and promptly resolves healing of a wound [38]. Several inflammatory cytokines active during immune response, such as IL-8, MCP-1 and VEGF, induce migration of endothelial stem cell lineage for new vessel formation. On the other hand, collagen itself serves as a promoter of angiogenesis. It has been reported that collagen type I has a potent influence of angiogenesis via integrin pathway in vitro and in vivo [39,40]. Moreover, C-peptide fragment of type I collagen is able to recruit endothelial cells in triggering neovascularization during wound healing. Simultaneously, new blood vessels facilitate secretion of multiple cytokines such as TNF-α and IL-6, and growth factors such as TGF-1β and IGF-1, leading to a promotion of differentiation of MSC to fibroblast. Differentiated fibroblasts participate in synthesizing collagen and remodeling of ECM through the regulation MMPs and TIMPs [1,40].

### 3.2. Guided Bone Regeneration:(GBR)

Guided bone regeneration (GBR) is an established technique extensively used for alveolar ridge augmentation [41]. GBR requires bone substitutes or implants to fill the defect and needs a barrier membrane to maintain space in the bone defect, creating a favorable microenvironment for repopulation of bone cells [18,41]. Ideally, membranes for GBR should prevent non-osseous cells ingrowth and be able to facilitate growth of osseous cells, vascularization of healing tissue, and bone regeneration [42]. Non-resorbable and resorbable membranes are two main types used for GBR. Non-resorbable membranes, such as polytetrafluoroethylene (PTFE), expanded PTFE (e-PTFE), titanium (Ti), and ti-reinforced PTFE (Ti-PTFE), eventually need post-operative reopening of the soft tissue for removal of the membranes, a process that is not required for the resorbable membranes [43,44]. Collagen is biodegradable and also facilitates proliferation and differentiation of osteoblasts for osseous regeneration. Collagen membranes are used for guided bone regeneration, acting as an inhibitory barrier against fibroblast or soft tissue ingrowth to allow a suitable niche for bone regeneration, particularly in alveolar bone and ridge defects. Several types of collagen membranes have been developed in combination with several biopolymers to optimize treatment strategies. Collagen membranes show very different ultrastructures, which refer to their physio-chemical and mechanical properties with different biological functions. From the literature, natural biopolymers with a porous micro-scale are more suitable for angiogenesis, nutrient transport, and mineral deposit for bone formation [45,46]. In contrast, nano-scale membranes are excellent for cell attachment and differentiation, and being inhibitory barriers against physical damages and pathogens to maintain the microenvironmental niche for bone regeneration [47,48]. Different ultrastructures also affect their mechanical and chemical properties; a more compact and denser ultrastructure shows a higher time for chemical degradation as indicated by trypsin degradation of a dense structure of multi-directional networks in a Jason^®^ membrane compared to a network of collagen fiber bundles of a Biocollagen^®^ membrane. Moreover, a higher stiffness and Young modulus value were also presented in the denser and more compact ultrastructure of a Jason^®^ membrane compared to a Biocollagen^®^ membrane [46]. Each type of membrane offers both advantages and drawbacks; dentists therefore need to make a decision based on the understanding of treatment strategies and the properties of membrane products.

Plenty of collagen-based membranes are available in the market and have been extensively used in GBR for many clinical situations, including periodontal osseous defects, alveolar bone augmentation, dental implantation, and peri-implantitis. Using CelGro^TM^, a non-cross-linked type I collagen-based membrane, prevents soft tissue ingrowth and provides a suitable microenvironment to activate ECM of bone and new bone formation. GBR treatment with CelGro^TM^ successfully regenerates new mature bone to stabilize dental implants and crown placement [18]. Moreover, analysis of studies related to use of GBR with collagen membrane in implants showed that this procedure had a high implant survival rate of 98.34% with reduced risk of morbidity and healing time, corrected alveolar ridge, and increased patient comfort [41]. Thus, the presence of collagen membranes and their bioactive functions provides higher levels of bone regeneration and restitution of the defected bone site compared to treatment of the defect without a barrier membrane.

Mechanistically, when collagen membranes are placed, they mainly act as a physical barrier in the prevention of cell ingrowth from surrounding tissue, to create and allow a suitable microenvironment for complete regeneration in the area of lesion [49,50,51]. In its biological participation, it also facilitates the recruitment of CD68-positive monocyte/macrophage cells and periostin-positive osteoprogenitors from surrounding tissues to adhere to the membranes. These cells produce and secrete a variety of pivotal factors such as fibroblast-growth factor 2 (FGF-2), TGF-b, BMP2, RANKL, VEGF, and others. FGF-2 and TGF-b mainly function in regulation of non-osseous tissue remodeling through promoting fibroblast proliferation, stimulating collagen synthesis, and modulating activity of MMPs and TIMPs [42,52]. On the other hand, BMP2 and RANKL activate activity of osteoblasts and osteoclasts, respectively, leading to development of new osseous tissue via bone formation and remodeling processes [53,54]. The presence of VEGF contributes to create new vascularization in the defected area by triggering endothelial lineage migration, proliferation, and differentiation to form new blood vessels, and further maintaining several needed signal molecules for tissue and bone remodeling [42,55].

### 3.3. Being Carrier and Transport Vehicle of Collagen in Scaffold/Matrix Forms

Collagen is used as carrier or transport vehicle for osteoconductive effect in the form of scaffolds in combination with bone graft materials and substitutes such as demineralized bone matrix [23,24,56,57]. In addition, collagen is also applied as a scaffold/matrix for cell and gene therapies, because it has potential for biocompatibility, degradability, low antigenicity, and cell adhesion in providing three-dimensional distribution [3,58,59]. 

Regarding to the mechanism of action, collagen acts as an extracellular matrix in providing three-dimensional distribution for cell adhesion. The cells, particularly bone cells for bone graft/substitution or stem cells for cell regenerative therapy, have been proposed to bind with collagen through collagen-binding receptors. Integrins are the major family of cell adhesion receptors involved in cell-collagen binding [60]. In addition to integrins, discoidin domain receptors 1 and 2 (DDR1 and DDR2), leukocyte-associatedimmunoglobulin-likereceptor-1(LAIR-1), and annexin V or anchorin II have also been implicated in cell-collagen binding [49,61,62]. In addition to these receptors, collagen also has receptor-binding sites for cell-binding collagen. Four different sites have been reported; (1) the specific motifs inside triple helix of collagen which represent highly specialized affinity, (2) a common sequence of collagen such as GPO sequence (G: glycine; P: proline; O: hydroxyproline), (3) a cryptic motif of collagen that can be recognized only after being denatured, and (4) a non-collagenous domain (NC) like NC1 domain [60]. Thus, cellular adhesive response to collagen products may depend on the dominance of expressed receptor types on plasma membrane of the cells, and the number and type receptor-binding sites of collagen.

#### 3.3.1. Bone Graft and Substitution

Collagen is used as carrier or transport vehicle for bone graft and as substitution for the regeneration of periodontal tissue in order to augment and reconstruct periodontal and alveolar bone defects. In clinical trials, SmartBone^TM^, a common bone graft material comprised of demineralized bone matrix combined with collagen, was utilized in periodontal osseous defect and alveolar ridge augmentation. The data exhibited that after four months of placement, successful osseointegration and new bone formation appeared with vascular connective tissue surrounding it, and new bone was formed with increased alveolar bone dimension [23,24]. However, the poor mechanical and high biodegradable properties of collagen scaffolds limit its application to some extent cases [63]. To overcome these, collagen scaffolds are thus modified by some physical methods or cross-linked by some chemicals and natural/synthetic polymers. For example, a fabricated collagen-chitosan scaffold by polyelectrolyte complexation and a blended collagen with natural protein polymer silk fibroin is beneficial in increasing mechanical properties and lengthening biodegradability of collagen [63,64]. Furthermore, since collagen itself provides an osteoinductive effect, it therefore can be applied as an alternative to allograft-derived materials that are available in the market. It has been reported that these materials have a potential in being a suitable environment for bone mineralization, neovascularization, and growth factor adhesion [65]. However, they are less considered for graft substitution as they exhibit high risk of adverse immune responses and possess poor structural integrity, especially in the long-term application of grafting case studies [50,66].

#### 3.3.2. Gene and Cell Regenerative Therapies

Cell therapy can be described as treatment that introduces new cells into a tissue whereas gene therapy is defined by the use of genetically modified cells to deliver a specific gene of bioactive proteins [3,51]. Collagen products and collagen scaffold/matrix have been utilized in these two therapies to provide structural and biochemical stabilities for newly proper tissue formation [67]. Using collagen as matrix for recombinant adenoviral vectors encoding BMP-7 (Ad-BMP7) in bone tissue engineering around titanium dental implants resulted in sustaining expression for up to 10 days. Treatment of Ad-BMP7 enhanced filling of alveolar bone defects, new bone formation, and new bone-to-implant contact [16]. Furthermore, vector distribution of Ad-encoding platelet-derived growth factor B (PDGF-B) delivered in collagen matrix was localized in the defected area without distant organ involvement. No significant difference of histological evaluation, specific hematological, or blood chemistries were found during investigation [17]. 

Likewise, a study of periodontal ligament stem cells (PDLSCs) attached to a collagen scaffold revealed extensive coverage of new alveolar bone in ovine and rat periodontal defect models. In human clinical investigation, placement of dental pulp-MSC in a collagen sponge scaffold (Gingistat^®^) could repair bone defects, following one year of grafting. Histological observation clearly revealed the complete regeneration of alveolar bone at the area of lesion [15]. Therefore, collagen as a delivery system in matrix form is considered safe for possible application in humans for repair and/or regeneration in terms of gene and cell therapies.

### 3.4. Limitations of Collagen-Based Biomaterials in Periodontal Regeneration

Although those customized collagen products attempted to regenerate periodontal tissues in proper structures, some regenerating functional periodontal tissues are still limited as follow: (1) the fine structures, called Sharpey’s fibers, are not restored, resulting in unstable connection between cementum, PDL, and alveolar bone. With unstable connection, regenerated periodontal tissues incompletely support teeth and bear occlusal force; (2) horizontal alveolar bone loss cannot be completely restored; and (3) long-term stability of regenerated periodontal tissues is commonly reduced. Therefore, novel materials and customized methods that can closely mimic the architectural hierarchy of periodontal tissues are prerequisite to achieve periodontal tissue regeneration, both structurally and functionally.

## 4. Plant-Based Collagen and Its Perspective

### 4.1. History

Although collagen used for pharmacological and biomedical applications is historically extracted from animal sources, substantiated concerns have been raised with the risk of infectious disease transmission known as bovine spongiform encephalopathy and biophysical profile of extracted collagen on post-depositional modification [68,69]. Evidence of immunogenic studies also reported that 2–4% of treated population is allergic to bovine and porcine-derived collagen by evoking both cellular and humoral immune responses [69]. Marine-derived collagen, which is successfully extracted form fish skin, scales, and bones, has recently become of interest because of a plentiful source of collagen-containing tissues from the marine food industry and an absence of disease transmission [69]. Biological demonstrations of marine collagen revealed high biodegradability, high biocompatibility, and low immunogenicity [69,70]. However, chemical and mechanical characterizations of marine-derived collagen have been reported to be different from animal collagen in various properties, including lower water solubility, lower melting point, lower viscosity, higher fibrillar proportion of glutamate and alanine, and lower proportion of proline and hydroxyproline contents [69,71,72]. With the noted differences, marine collagens are not excluded from biotechnological use; collagen with a low melting point, low solubility, or others can be modified by additional crosslinking to be suitable for biomaterial and bioengineering applications.

In addition to animal and marine sources, there is a growth in technology producing human collagen from plants through recombinant expression system [73,74,75]. Tobacco plant-based expression of rhCOL1, which drives simultaneous vacuole-targeted expression and post-translation of type I collagen, provides a new source of collagen and can be applied for tissue engineering and regenerative medicines [75]. This collagen is homogenic and has the ability to self-assemble, forming homogenous fibrils that display intact binding sites with other combined materials or bioactive agents [68] (Table 2). In the future, the shortcomings of epidemic diseases, religion restriction, and environmental concerns in using traditional collagen sources might be overcome and substituted by plant-derived collagen production.

### 4.2. Plant Recombinant Human Collage

Recombinant expression systems have emerged as a potentially alternative synthetic method to produce exogenous human collagen, providing homogenic collagen fibrils without the risk of pathologic infection [73,74,75]. Recombinant expression in prokaryotic and yeast systems were first attempted; however, the systems result in relatively low stability and short fibers. These are likely due to the lack of necessary enzymatic support to produce post-translationally modified collagen [69].

Several plant recombinant systems have been used to express and produce plant-extracted rhCOL1. Expression of recombinant proteins in plant is generated by sorting and targeting recombinant gene expression of the interest proteins to subcellular compartments such as nucleus, cytosol, and chloroplast. A relatively high output system of rhCOL1 has been reported in the tobacco plant (*Nicotiana tabacum*) [68,69,75]. In this system, two human genes encoding recombinant type I collagen and enzymatic human P4H-a, P4H-b and LH3 genes are targeted and expressed in vacuoles of tobacco plant. Vacuole-targeted expression of five genes can provide procollagen yielding at 1 g/kg of dry tobacco leaves. The isolated protein is decorated by a proportion of hydroxylated proline and lysine residuals (7–10% and 0.7–1%, respectively), and exhibits fibril-forming capacity, thermostability in triple helical structure, and resistance to protease activity up to 39 °C, similar to native human collagen [69,75,77]. Biological properties of the resulting proteins are also similar to human collagen in terms of supporting cellular adhesion and expansion of human cell during tissue repair or regeneration. Furthermore, the extracted protein is highly hydrophilic [68,75], indicating that collagen made from plant recombinant system has physicochemical characterizations identical to those of its natural human collagen without concern of immunogenicity and disease transmission. Hence, plant-derived rhCOL1 may be ideal as biomaterial for biomedical use.

### 4.3. Medical Studies of Using Plant Recombinant Human Collagen

The homogenous plant-derived rhCOL1 created by vacuole-targeted expression has been investigated as a reliable source material for regeneration. Scaffolding properties of plant-extracted rhCOL1 were determined through electrospinning and freeze-drying in comparison to bovine collagen. Both raw materials could be formed into two common scaffold types, the electrospun nonwoven scaffold and lyophilized sponge, with similar structures. H&E-stained histological sections had no marked differences in skin morphology between engineered skin fabricated with plant-derived rhCOL1 and bovine collagen [81]. Plant-extracted rhCOL1 was safe in using as drug delivery scaffold material. Injection of Arthrex ACP^®^ Tendo (ACP) combined with plant-derived rhCOL1 enabled prolonged release of ACP to the injury site with only one injection [87]. Significantly, plant-derived rhCOL1 supported endothelial, fibroblast, and keratinocyte cell attachment and proliferation at a higher level than a commercial scaffold made of bovine collagen. In addition, activated THP-1 macrophage had approximately two-fold lower IL-1β secretion when exposed to electrospun plant-derived rhCOL1 scaffold compared to the bovine collagen scaffold [81]. Plant-extracted rhCOL1 also showed no immune response in a sensitive assay with human CD4^+^ T cells, while a higher response was observed in bovine collagen [80].

Shilo et al. demonstrated the ability for wound healing of plant-derived rhCOL1 gel in a rat cutaneous wound model and found that plant-derived rhCOL1 induced a more rapid healing process. Within 21 days, 95% wound closure was observed with enhanced reepithelization and reduced inflammation, whereas 68% closure of wound was achieved by bovine collagen products [84]. Clinical investigation also showed a significantly shorter healing time related to plant-extracted rhCOL1 gel (64 ± 4 days versus 90 ± 11 days) with a significantly greater wound size reduction (78% vs. 50%). In addition, there was no local or systemic adverse events or hypersensitivity for the plant-derived rhCOL1 used [88].

The regenerative potential of plant-derived rhCOL1 was assessed in hydrogel form. Crosslinking the plant-extracted rhCOL1 with 1-ethyl-3-(3-dimethyl aminopropyl) carbodiimide and N-hydroxysuccinimide produces a robust and transparent hydrogel without having an effect on the outcome of biocompatibility testing. Histological data showed that stratified corneal epithelium and mucin-positive tear film were fully differentiated and regenerated, respectively [85]. Biomaterials made from plant-derived rhCOL1 are biocompatible, cause zero immune response, are mechanically stable, and promote regeneration of damaged tissues [80,81,84,85,87,88]. Therefore, plant-derived rhCOL1 might eventually serve as a viable source of collagen for regeneration in tissue engineering and regenerative medicine applications.

### 4.4. Perspective in Periodontal Treatment

There are no studies published on the effects of using plant-derived rhCOL1 as biomaterials for regenerative medicine in periodontal diseases, as well as dental diseases. However, investigating characterization of plant-based collagen showed a pure heterotrimeric type I collagen with a ratio of hydroxyproline and lysine contents similar to native human collagen and a preserved α-helical structure. Moreover, it presents D-band striations that play an implicating role in mechanical properties and bio-functionality with high levels of integrin binding sites [68,75]. From a biological perspective, plant-extracted rhCOL1 supports the attachment and proliferation of several cell types, such as endothelial cells, fibroblasts, and keratinocytes [81]. It also promotes rapid wound healing processes by enhancing reepithelization with fully regenerated and differentiated tissues and cells in the damaged site [84,88]. In addition, it is considerably determined to be a safe alternative to allograft, as indicated by a lower secreted IL-1β level than bovine collagen in activated THP-1 macrophage model and no cellular response in human CD4^+^ T cells sensitive assays [80,81] (Figure 3). Therefore, it is possible that this new technology may work toward a common goal of achieving functional periodontal tissue regeneration with low-cost, effective in extraction and purification of collagen in comparison to animal sources, and avoidance of epidemic diseases, immunogenic adverse effects, religion restriction, and environmental concerns related to using animal collagen. However, future research is necessary to confirm this possibility.

## 5. Summary and Future Directions

There is a broad spectrum of commercially available collagen-based biomaterials for periodontal regeneration in several forms, including collagen sponges, plugs, scaffolds, matrices, and nanoparticles incorporating bioactive molecules or stem cells. The issue of collagen sources is one which persists beyond the scope of this article. Animal-based collagen possess substantiated risks for disease transmission and immunogenicity. The coming of recombinant DNA technology in producing rhCOL1 in tobacco plants might overcome and substitute the use of collagen from animal sources, because rhCOL1 provides a pure heterotrimeric type I collagen with similar architecture and biological properties to native human collagen. In addition, rhCOL1 avoids pathogenic contamination and transmission, immunogenic adverse effects, religion restriction, and environmental concerns related to using animal collagen. However, future studies are needed to confirm the periodontal regenerative potential of rhCOL1.

## Figures and Tables

**Figure 1 biomimetics-07-00034-f001:**
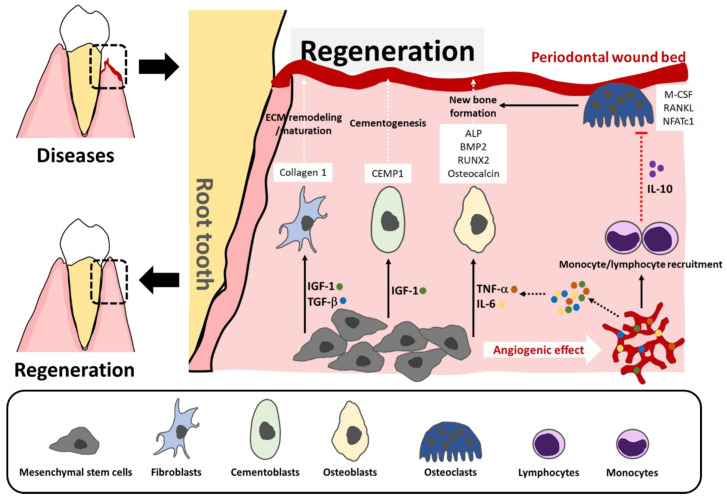
Mechanisms of periodontal regeneration. MSC at the area of lesions exerts angiogenic effects, resulting in the secretion of inflammatory cytokines (TNF-a and IL-6) and growth factors (IGF-1 and TGF-b), and recruitment of immune cells (monocytes and lymphocytes). These inflammatory mediators induce proliferation and differentiation of MSC towards fibroblasts, cementoblasts, and osteoblasts leading to ECM remodeling, cementogenesis, and new bone formation, respectively. The immune cells release IL-10 to suppress osteoclast activation in homeostatic maintenance of new bone formation.

**Figure 2 biomimetics-07-00034-f002:**
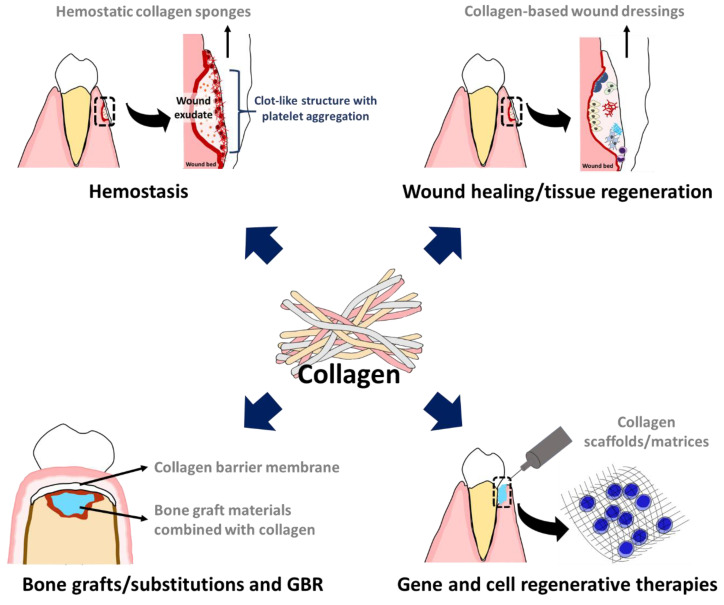
Application of collagen as biomaterials for periodontal regeneration. Collagen is commonly used as hemostatic materials, wound dressing products for healing and regeneration, bone grafting materials for bone substitution and GBR, and scaffolds/matrices for gene and cell regeneration therapies.

**Figure 3 biomimetics-07-00034-f003:**
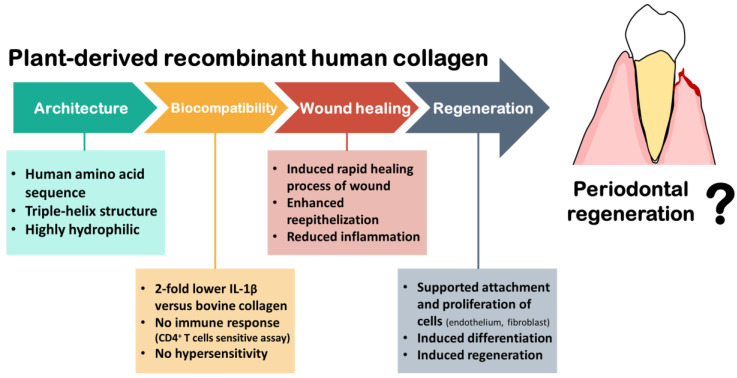
Chemical and biological properties of plant-derived recombinant human collagen and its perspectives in periodontal regeneration.

**Table 1 biomimetics-07-00034-t001:** Studies of collagen applications and outcomes in periodontal regeneration.

Therapies	Collagen Applications	Outcomes	References
**Gene and cell regenerative therapies**	Collagen hydrogel scaffold (prepare from atelocollagen: type I collagen) as scaffold/matrix for recombinant human BMP2	Reconstruction of cementum-like tissue, periodontal ligament and alveolar bone, and prevention of ankylosis in one wall intra-bony defect	Kato et al., 2015 [14]
	Gingistat^®^ collagen sponge scaffold with dental pulp stem cells (DPCs)	Optimal vertical repair of alveolar bone, and complete restoration back to the second molar of periodontal tissue in the model of injury site by extraction of mandibular third extraction	d’Aquina et al., 2009 [15]
	2.6% collagen as matrix for Ad-BMP7 gene delivery	Promotion of alveolar bone defect, and enhancement of new bone-to-implant contacts in edentulous ridge defect followed by dental implantation	Dunn et al., 2005 [16]
	2.6% collagen matrix containing Ad-PDGF-B gene delivery	Increase in bridging bone and tooth-lining cemental regeneration in periodontal defects (large tooth-associated alveolar bone defect)	Jin et al., 2004 [17]
**Guided bone regeneration**	Porcine-derived collagen barrier membrane (CelGro^TM^, Bio-Gide^®^)	Restoration of bone defect in both horizontal and vertical dimension, and sufficient support to the implants with no adverse effects in the GBR for dental implants	Allan et al., 2021 [18]
	AlloDerm^®^ GBR as barrier membrane with bone grafting materials (BioOss)	Significant induction of ridge growth in both horizontal and vertical dimension in soft and hard tissues in class I ridge defect	Sabitha Sudarsan et al., 2008 [19]
	Colla-Guide resorbable barrier membrane in combination with Osteon	Preservation of alveolar crest shape and height by restoring bone tissue via secondary osseous tissue formation in maxillary alveolar process fracture	Boymuradov and Shukhrat, 2011 [20]
	Creos^TM^ Xenoprotect in combination with bone grafting materials (BioOss)	Enhancement of bone augmentation of alveolar ridge with accelerated healing time in horizontal alveolar ridge defects	Wessing et al., 2016 [21]
**Bone graft and substitution**	BioMend^®^ type I bovine collagen	Serving as barrier in the prevention of epithelial cell migration/invasion in allowing GBR and GTR regenerations	Sheikh et al., 2017 [22]
	Demineralized bone combined with collagen matrix (SmartBone^TM^)	Successful osseointegration and new bone formation appeared with vascular connective tissue surrounding periodontal osseous defect	Mandelli et al., 2018, Abuelnaga et al., 2018 [23,24]
	Mineralized collagen bone grafting material	Stimulation of new bone formation to reconstruct the deficient alveolar ridge around the dental implant	Wang et al., 2019 [25]
**Hemostasis and wound healing**	CollaPlug^®^ Bovine-derived collagen	Exerting effective local hemostasis, acceleration of healing in soft tissues, and reduction of post-operative pain	Abdelaziz et al., 2015 [26]
	Bovine-derived collagen membrane (Dressing products from EUCARE pharmaceuticals)	Increase in the scores of hemostasis, granulation tissue formation, and epithelization with reduced pain score in various intra-oral lesions	Sowjanya et al., 2016 [27]
	Absorbable type I collagen sponge (Ateoplug)	Enhancement of tissue regeneration by promoting proliferation and differentiation of MSCs in periodontal tissue, facilitation of endogenous healing of wound, and prevention of post-operative complications in third molar extraction socket	Cho et al., 2015 [28]
	Bio-resorbable type I bovine collagen (Healiguide^TM^)	Significant reduction in gingival recession defects with higher clinical assessment values (recession depth, root coverage percentage, probing depth, clinical attachment level, gingival tissue thickness, and others)	Mahajan et al., 2018 [29]

Ad-BMP7: adenovirus containing bone morphogenic protein 7, Ad-PDGF-B: adenovirus containing platelet-derived growth factor-B, BMP2: bone morphogenic protein2, GBR: guided bone regeneration, MSCs: mesenchymal stem cells.

**Table 2 biomimetics-07-00034-t002:** Comparison of current sources of medical collagen.

Features	Animal Collagen	Marine Collagen	Plant Collagen
**Architecture/structure**	Highly homologous to human collagen (component of amino acid sequences depend on source of extraction (species and body part) Triple helix structure [76]	Highly homologous to human collagen (but lower proportion of Pro and Hyp contents) Triple helix structure Lower water solubility, melting point, and viscosity (since lower Pro and Hyp) [71,72]	Human amino acid sequence (7–10% and 0.7–1% of hydroxylated proline and lysine, respectively) Triple helix structure [68,75,77]
**Biocompatibility**	Around 2–4% of treated population is allergic to bovine and porcine-derived collagen, evoking both cellular and humoral immune responses [69] Risk of infectious disease transmission, such as bovine spongiform encephalopathy [68,69]	Biocompatibility meets the standards of FDA on the sub-chronic toxicity test (ISO-10993) [78] Less immunogenic with low inducing pro-inflammatory cytokines as well as inducible NO synthase [79]	Free of foreign animal tissue contamination No immune response (CD4+ T cell sensitive assay) [80] No hypersensitivity Two-fold lower IL-1b level vs. bovine collagen [81]
**Regenerative ability**	Enhancing tissue regeneration by promoting proliferation and differentiation [28] Facilitating endogenous wound healing with prevented post-operative complications [28] Providing sufficient support for tissue and cell regeneration [18]	Enhancing cell and tissue regeneration [82] Supporting cell adhesion and inducing proliferation and differentiation [83] Promoting healing of wound by accelerated endogenous healing process [82]	Supporting cellular adhesion and expansion of human cell during tissue repair or regeneration [81] Enhanced reepithelization [84] Histological data showed that the damaged tissues were fully differentiated and regenerated [85]
**Restriction**	Religion restriction	No religion restriction	No religion restriction
**%Yield**	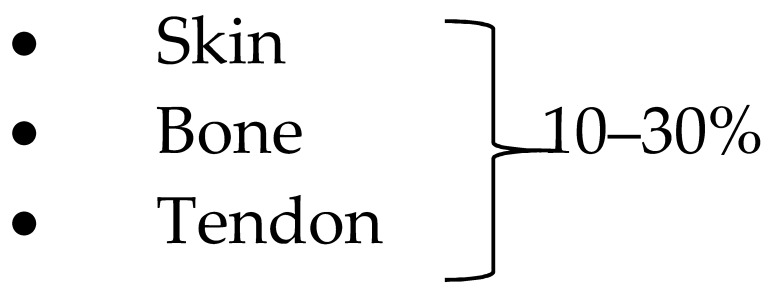 Depending on the species, sex, age and body weight of the animal [86]	Skin (20–90%) Scale (0.05–3.2%) Depending on species and extraction conditions [70]	1 g/kg (0.1%) [75]

## Data Availability

Not applicable.

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
