# Peer review of "Collagen-Based Biomaterials in Periodontal Regeneration: Current Applications and Future Perspectives of Plant-Based Collagen"

_biomimetics, 2022, doi:10.3390/biomimetics7020034_

Round 1
Reviewer 1 Report
This review provides the current progresses of periodontal regeneration. Specifically, it was focused on current trends in preclinical, clinical research and the potential for clinical translatability in using collagen products as biomedical materials for hemostasis, wound healing, bone substitution and guided bone regeneration in periodontics. In addition, special attention in cellular mechanisms of collagen utilization as scaffolds/matrices for gene and cell regenerative therapies is presented. The manuscript needs minor revision.
The manuscript needs to check the English writing.
The new references in the field of the manuscript should be added especially from 2021 and 2022. For example, important references from new articles PMCID: PMC7584591, PMID: 34931969
Reviewer 2 Report
This paper was intended to review the collagen application in the periodontal regenerative therapy, and it indeed had briefly introduced some basic concepts regarding this topic. However, the entire article digressed from the main subject and did not seems to bring significant contribution to the field as the following reasons:
First, the detailed review of the mechanisms of collagen application specifically for periodontal regenerative therapy was not provided. This is the foundation of the advantages of collagen as a promising products to be further improved.
Second, the clinical or preclinical status of the collagen products using the periodontal regenerative therapy should be clearly introduced for the readers to know how is the treatment outcome and what is to be improved.
Third, the research trend should be pointed out.
Overall, this article has provided indistinct information and no updates in this field.
Reviewer 3 Report
Please see the attachment.

Round 2
Reviewer 3 Report
The authors significantly improved their manuscript by addressing all the comments and suggestions. The actual version of the manuscript is ready to be published.